# Updating Mothers within an Hour of Newborn's Admission to Neonatal ICU

Shabih Manzar 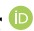

Section of Neonatology, Department of Pediatrics, Louisiana State University Health Sciences Center, Shreveport, LA 71103, USA; shabih.manzar@lsuhs.edu; Tel.: +1-318-626-1623

**Abstract:** Background: Patient satisfaction is tightly linked with healthcare quality and high-value care. Timely communication is important in attaining patient satisfaction. The aim of the study was to provide all delivering mothers an update within an hour of their newborn's admission to the neonatal intensive care unit (NICU). Methods: An educational module was developed with a PowerPoint presentation on the role of a timely update. The team, consisting of the neonatal nurse practitioners (NNP) and residents, were provided access to the presentation. After completing the presentation, they completed a questionnaire showing understanding. The principal investigator met with the mother after admissions to assess if she was updated within an hour of the admission of her baby to the NICU. Results: A total of 22 mothers participated in the study. Thirty-six percent of the mothers were updated within an hour of admission of their neonates to the NICU. The average time taken from admission to updating mothers was $5.75 \pm 6.7$ h. All mothers were satisfied with the explanation of the NICU staff. Conclusions: We noted a 100% satisfaction rate; however, we observed a low percentage of communication with the mother. The study provided the baseline data for the next PDSA cycle.

**Keywords:** PDSA cycle; communication; healthcare quality; patients' satisfaction

## 1. Introduction

Preterm birth is associated with increased stress on the parents, which can influence bonding. Mothers were reported to have increased stress after preterm birth and admission of their infants to neonatal intensive care (NICU) [1,2]. According to a review by Tahirkheli [3], the mothers of infants admitted to the NICU are at greater risk for relationship difficulties, family stress, and financial stress. They are 40% more likely to develop postpartum depression (PPD). Depression is reported to be the most common psychological disorder during the perinatal period [4]. The depression is highest during the admission phase of NICU and then decreases over time [1]. It is therefore very important to involve parents as soon as possible in care and communication to facilitate family-centered care [5].

Patient satisfaction is used as a quality indicator in health care. Poor communication from healthcare providers and lack of empathy leads to dissatisfaction [6]. Therefore, it is very important to provide timely update to the mothers (parents) about their babies admitted to the NICU.

This study was conducted with the principal objective of improving healthcare quality by improving patient satisfaction. The rationale and specific aims were addressed per the SQUIRE 2.0 guidelines [7,8]. We implemented a practice change of updating mothers within an hour of their newborns' admission to the NICU through education of staff. We used the quality improvement framework of the Institute for Healthcare Improvement (IHI) [9] method of improvement, using the plan-do-study-act cycle (what are we trying to accomplish—patient satisfaction through early updates, how will we know that a change is an improvement–by patients' responses to a questionnaire, what change can we make that will result in improvement—educate the staff on the value of early communication with

the patient). The SMART AIM included a specific objective (to update 50% of the mothers within an hour of admission), measurable data (number of updates/total admissions), actionable process change (update rates), realistic change (maternal satisfaction), and finally, a timely project (one month). The driver diagram is shown in Figure 1.

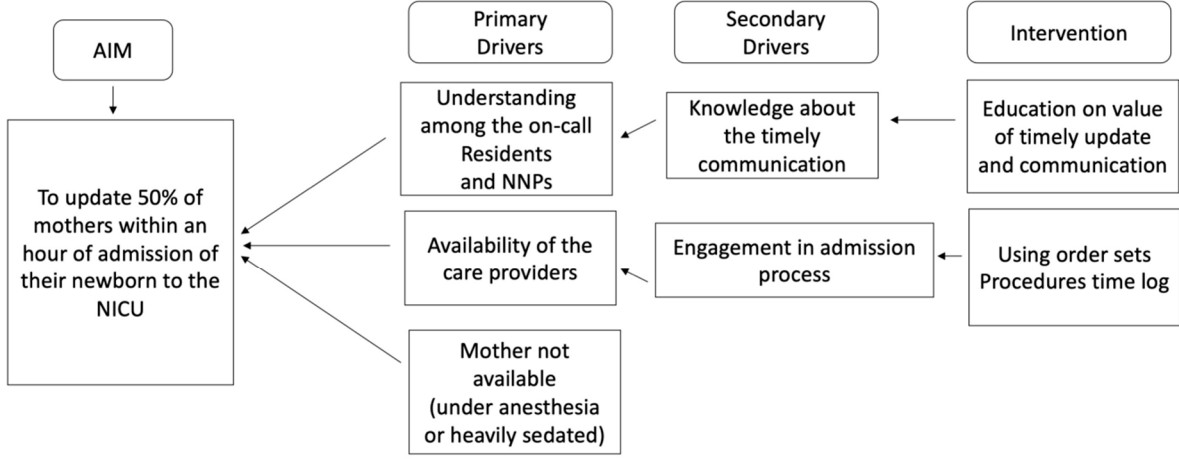

NICU – Neonatal Intensive Care Unit
NNP – Neonatal Nurse Practitioner

**Figure 1.** Driver diagram.

## 2. Methods

This study was approved by the Institutional Review Board of our institution. The protocol number was STUDY00002051. The need for consent was waived, as data were collected as questionnaires. The study consisted of the first cycle of the plan-do-study-act. The care team involved in the NICU admission process consisted of neonatal nurse practitioners (NNP) and residents rotating through the NICU. An educational module was developed using a PowerPoint presentation on the role of timely update (Supplementary file) and the value of communication based on the earlier report [10,11]. The educational material was distributed to the residents and NNPs via institutional email. They reviewed the text in their personal time, then completed and submitted the questionnaire. It took 3 days for the whole group to complete the task. The principal investigator met with the mother after admissions to assess if she was updated within an hour of the admission of her baby to the NICU. The update was documented by the NNP or resident on a data collection form.

## 3. Results

A total of 22 mothers participated in the study during the study period of one month. Thirty-six percent of the mothers were updated within an hour of the admission of their neonates to the NICU (Figure 2). The average time taken from admission to updating mothers was 5.75 h (Table 1). It was alarming to note that only 20% of the residents and NNPs appreciated postpartum depression as a common problem (Figure 3). On further analysis, we noted that less than 20% of the mothers were under sedation, meaning they were alert to communication within an hour. It was reassuring to note that 100% of mothers were satisfied with the explanation from the NICU team member (Figure 4).

**Table 1.** Summary of the Update Times.

| Case Number | Time of Delivery | Time of First Contact | How Many Hours after Admission the Mom Was Updated |
|:---:|:---:|:---:|:---:|
| 1 | 15:48 | 8:15 | 16:27 |

**Table 1.** *Cont.*

| Case Number | Time of Delivery | Time of First Contact | How Many Hours after Admission the Mom Was Updated |
|:---:|:---:|:---:|:---:|
| 2 | 15:36 | 9:00 | 17:24 |
| 3 | 2:38 | 3:15 | 0:37 |
| 4 | 17:23 | 8:05 | 14:42 |
| 5 | 13:40 | 14:20 | 0:40 |
| 6 | 11:13 | 14:45 | 3:32 |
| 7 | 7:30 | 8:15 | 0:45 |
| 8 | 13:33 | 8:05 | 18:32 |
| 9 | 9:00 | 9:30 | 0:30 |
| 10 | 18:05 | 8:45 | 14:40 |
| 11 | 13:20 | 13:55 | 0:35 |
| 12 | 13:24 | 15:20 | 1:56 |
| 13 | 7:53 | 10:15 | 2:22 |
| 14 | 16:15 | 18:45 | 2:30 |
| 15 | 4:10 | 5:05 | 0:55 |
| 16 | 5:45 | 7:50 | 2:05 |
| 17 | 10:46 | 11:20 | 0:34 |
| 18 | 5:13 | 6:05 | 0:52 |
| 19 | 5:57 | 10:40 | 4:43 |
| 20 | 8:15 | 12:15 | 4:00 |
| 21 | 2:42 | 10:20 | 7:38 |
| 22 | 17:50 | 9:05 | 15:15 |

Time taken from admission to update (hours): Mean = 5.75 h, standard deviation 6.7 h, Median = 2.2 h, Range = 0.3–18.32 h.

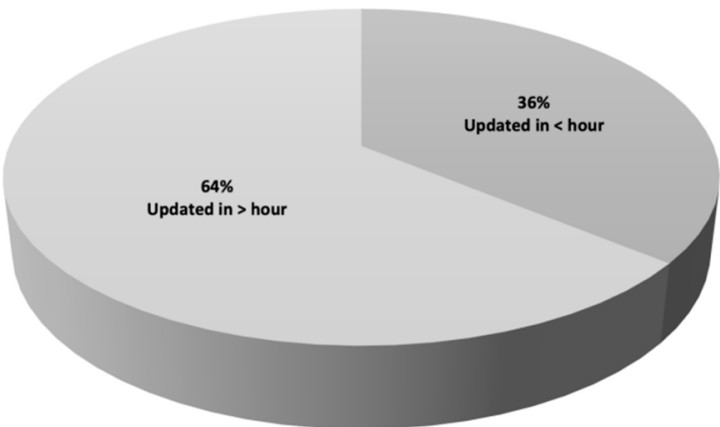

**Figure 2.** Distribution of update times (percentages).

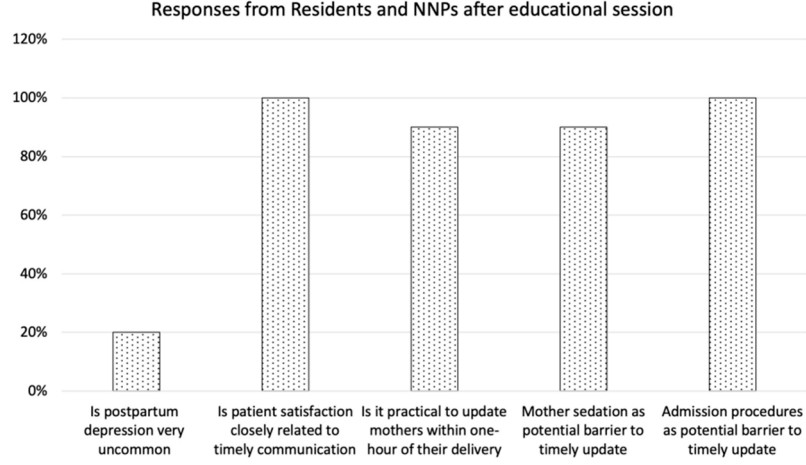

**Figure 3.** Summary of responses from residents and neonatal nurse practitioners after educational session.

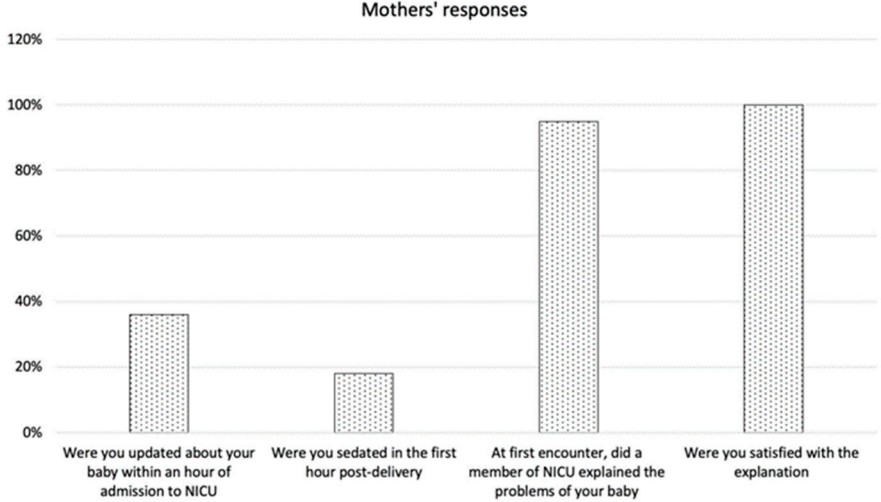

**Figure 4.** Summary of responses from mothers.

### 4. Limitations

Although the study did show good patient satisfaction, there was no relation to the timely update, which was used as a surrogate for satisfaction. In a previous study, Day et al. [12] described a lack of statistically significant difference in patient satisfaction and communication from nurses and physicians. The response to complications with care was presented as the reason for the discrepancy. The other limitation of our study was that the satisfaction question was not scaled or scored as an ordinal variable using a Likert response rather was a dichotomous option of yes or no. This may have affected the response rate. Staff shortage during certain shifts would have potentially affected the timely updates. We exclusively used the word "mothers" instead of "parents", as fathers were rarely present in the room to update. The study has novelty, as no previous study has been performed looking at updating mothers within an hour of their newborn's admission.

### 5. Discussion

In this study, we were able to demonstrate 100% satisfaction among the mothers whose newborns were admitted to the NICU. However, we were unable to achieve a 50% update rate within an hour, as we aimed. This has given us the opportunity to perform the second PDSA cycle in the following months to achieve the goal of 50%.

Our data show the need for more education of the staff on the risk of postpartum depression, as we noted a very low response. Once the staff appreciates the risk, they will take further action. Upon detailed analysis of the individual cases, the presence of an

attending physician at the time of admission was noted to be the most important factor in the timely updates. Evening and night admissions had longer update times.

The strength of the study is that it provides the baseline data on communication with mothers after their newborn babies are admitted to the NICU. A low percentage of timely updates will be addressed in the next PDSA cycle, following the phases and implementations of PDSA cycles as described earlier [13].

## 6. Conclusions

We were able to demonstrate 100% patient satisfaction; however, we were unable to achieve the aim of updating 50% of mothers within one hour of the admission of their newborn babies to the NICU. The second PDSA cycle is planned to achieve a higher update rate, thereby improving healthcare quality.

## 7. Implications

The main implication of our study is that it provides a baseline deficiency in one of the quality measures, timely communication. A follow-up PDSA cycle that aims to increase timely updates by 20% is planned. The focus will be on educating the NNPs and residents while providing additional support when they are short-staffed. The identified barriers in this first PDSA cycle will be addressed in the next cycle.

**Supplementary Materials:** The following supporting information can be downloaded at: https: //www.mdpi.com/article/10.3390/standards2040033/s1.

**Funding:** This research received no external funding.

**Data Availability Statement:** Not applicable.

**Acknowledgments:** I would like to thank the medical staff for their help with the educational questionnaire.

**Conflicts of Interest:** The author declares no conflict of interest.

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
