# Peer review of "Updating Mothers within an Hour of Newborn’s Admission to Neonatal ICU"

_standards, doi:10.3390/standards2040033_

Round 1
Reviewer 1 Report
Comments and Suggestions for Authors
The manuscript entitled “Updating mothers within an hour of newborn’s admission to neonatal ICU” has been reviewed. The author tries to prove that by updating mothers the condition of newborns admitted to NICU, patient satisfaction could be improved. The author developed an educational module to standardized the updating process, and applied a questionnaire to evaluate patient satisfaction. The results appeared to be quite promising. However, there are some problems that require further explanation.
1. It would be better to present more information about the educational module since the severity and diagnosis of the newborn admitted to NICU might be different.
2. Why choose 1 hour in the research? Since there were less than 50% of the mothers were informed within 1 hour, why not choose 2 or 3 hours? If the newborn was in critical condition, emergent treatment by pediatric doctor could be ongoing during the time period. Are there any reference or data about the appropriate time period?
3. It would be better to separate the mothers into different groups according to the severity of their babies and re-evaluate the results.
4. The questionnaire is too simple. It is very difficult to judge patient satisfaction by only 4 questions with yes/no as their answer.
5. I would suggest some interview with the mothers and collect some qualitative data to support the results and conclusion would be a better way to improve the data from questionnaire.
Further improvement about the research questionnaire and evaluation was required.
Author Response
Response to Reviewer 1:
- It would be better to present more information about the educational module since the severity and diagnosis of the newborn admitted to NICU might be different.
More education is conducted via using zoom due to covid situation.
- Why choose 1 hour in the research? Since there were less than 50% of the mothers were informed within 1 hour, why not choose 2 or 3 hours? If the newborn was in critical condition, emergent treatment by pediatric doctor could be ongoing during the time period. Are there any reference or data about the appropriate time period?
We choose 1 hour as an arbitrary way. Usually by the 1 hour the admission process is completed in the NICU. We called this Golden hour.
- It would be better to separate the mothers into different groups according to the severity of their babies and re-evaluate the results.
We can divide mother based on prematurity, type of diseases, mode of delivery but this being a preliminary study, we decided to include all.
- The questionnaire is too simple. It is very difficult to judge patient satisfaction by only 4 questions with yes/no as their answer.
We kept it simple because our target was just knowing if moms were updated.
- I would suggest some interview with the mothers and collect some qualitative data to support the results and conclusion would be a better way to improve the data from questionnaire
Collecting the qualitative data is a great suggestion but we don’t have enough staff to do it.
Reviewer 2 Report
Comments and Suggestions for Authors
ABSTRACT:
Study objective: It is not well defined and is not resolved in the conclusions.
"The aim of the study is to provide all delivering mothers an update within an hour of their newborn's admission to the neonatal intensive care unit (NICU)." (lines 9 and 10).
INTRODUCTION:
The authors point out another objective of the study, which is too broad and poorly defined "This study was conducted with the main objective of improving healthcare quality by improving patient satisfaction." (lines 39 and 40).
MATERIALS, METHODS, AND RESULTS
The authors expose only the descriptive results, there is no deeper analysis of them.
DISCUSSION AND CONCLUSIONS
The authors have identified the results correctly and pointed out the discussion in a coherent, clear and precise way. This section has improved considerably.
Lastly, I would like to point out that the idea of ​​studying patient satisfaction by giving information in a timely manner to mothers (parents) of the newborn ‘admission to the neonatal intensive care unit (NICU) is interesting, but to the paper presented depth in the investigation, presenting the data in a descriptive way and with an elementary statistical method.
Author Response
Response to Reviewer 2:
ABSTRACT:
Study objective: It is not well defined and is not resolved in the conclusions.
"The aim of the study is to provide all delivering mothers an update within an hour of their newborn's admission to the neonatal intensive care unit (NICU)." (lines 9 and 10).
It is stated that 36% of the mothers were updated within an hour (line 16)
INTRODUCTION:
The authors point out another objective of the study, which is too broad and poorly defined "This study was conducted with the main objective of improving healthcare quality by improving patient satisfaction." (lines 39 and 40).
I agree, it is a broad indirect way of assessing healthcare improvement by patient satisfaction through timely updates. By definition, high value health care depends upon patient satisfaction.
MATERIALS, METHODS, AND RESULTS
The authors expose only the descriptive results, there is no deeper analysis of them.
The deeper analysis was not the purpose of this preliminary study.
DISCUSSION AND CONCLUSIONS
The authors have identified the results correctly and pointed out the discussion in a coherent, clear and precise way. This section has improved considerably.
Lastly, I would like to point out that the idea of ​​studying patient satisfaction by giving information in a timely manner to mothers (parents) of the newborn ‘admission to the neonatal intensive care unit (NICU) is interesting, but to the paper presented depth in the investigation, presenting the data in a descriptive way and with an elementary statistical method.
As we mentioned before, this is a preliminary study. We have launched a PDSA cycle to assess the process in detail.
Reviewer 3 Report
Comments and Suggestions for Authors
This is an interesting and important subject. Unfortunately, this paper did not present it in the best way.
Typically, it would be better to have the second PDSA cycle information, especially that there was success with the first PDSA.
Introduction:
Line 36 should be leads
Line 36-37: you state that the timely update is important, however, how do you define timely and what literature is there to support your statement.
Also, is it important to provide an update or in a specific time period?
Line 42: how did you come up with 1 hour intervention?
Line 48: Please write out your full SMART aim.
You probably had more drivers, primary and secondary. Please discuss them.
Methods:
Line 60: Please speak a little more about education. How was it disseminated, how did the team have access to it? When did they have to do it? On their personal time or during work hours? How long was it and how long did they have to complete it.
Line 63: if the PI went to ask the mother if she was updated within an hour, how do you know she remembered talking to anyone and especially the time when she spoke to people. Mothers have a lot of difficulty remembering information right after delivery. Was there any documentation that someone spoke to the mother? Something objective to compare with her memory of the fact?
Results
Line 69: no need to use adjectives for the SD, significance, etc. The reader will be able to determine it from the numbers.
Line 71: you jumped to the result of the survey without talking at all in the methods about the details of the survey or discussing the results of other aspects of the survey
Limitations:
the way this section is described is as part of the discussion section. Limitation and discussion sections are not well presented. There is no discussion as it relates to the results of this team and comparing them to previous work done.
Author Response
Many thanks for your valuable comments. Please see my response:
Typically, it would be better to have the second PDSA cycle information, especially that there was success with the first PDSA. The study if followed by by 3 PDSA cycle- the data is presented in a meeting and submitted as a separate paper.
Introduction:
Line 36 should be leads – changed, highlighted in yellow
Line 36-37: you state that the timely update is important, however, how do you define timely and what literature is there to support your statement. changed, highlighted in yellow
Also, is it important to provide an update or in a specific time period? changed, highlighted in yellow
Line 42: how did you come up with 1 hour intervention? One hour was arbitrary, we used the concept of Golden hour of newborn care, which is the first hour of life
Line 48: Please write out your full SMART aim. changed, highlighted in yellow
You probably had more drivers, primary and secondary. Please discuss them. changed, highlighted in yellow
Methods:
Line 60: Please speak a little more about education. How was it disseminated, how did the team have access to it? When did they have to do it? On their personal time or during work hours? How long was it and how long did they have to complete it. changed, highlighted in yellow
Line 63: if the PI went to ask the mother if she was updated within an hour, how do you know she remembered talking to anyone and especially the time when she spoke to people. Mothers have a lot of difficulty remembering information right after delivery. Was there any documentation that someone spoke to the mother? Something objective to compare with her memory of the fact? changed, highlighted in yellow
Results
Line 69: no need to use adjectives for the SD, significance, etc. The reader will be able to determine it from the numbers. All accessary information is deleted
Line 71: you jumped to the result of the survey without talking at all in the methods about the details of the survey or discussing the results of other aspects of the survey changed, highlighted in yellow
Limitations:
the way this section is described is as part of the discussion section. Limitation and discussion sections are not well presented. There is no discussion as it relates to the results of this team and comparing them to previous work done. changed, highlighted in yellow
Round 2
Reviewer 1 Report
Comments and Suggestions for Authors
The author has answered some of reviewer’s previous questions. The mothers of patients have been divided into groups, but not quite matched the requirement of reviewer. Reviewer has mentioned about the improvement of questionnaire and interview. Since the parts of questionnaire and qualitative data are unable to improve, I would consider that this manuscript is not qualified to be published.
Reviewer 2 Report
Comments and Suggestions for Authors
ABSTRACT
The research topic is of scientific and social interest.
There is cohesion in the article, between the section on Theoretical Framework and the subsequent sections, which describe the study and draw conclusions, have a correct, well-structured and cohesive design.
In general, the article is correct and I consider that the topic is in line with the journal’s research objectives.
INTRODUCTION:
The study objective is well defined and identified in both the abstract and the introduction.
The subject under investigation is of growing scientific and social interest. The investigation is current.
The result is a work that can be the basis for many others in this field.
The statistical technique used is well justified and explained, both the process and the results, despite not being widely used in these investigations, which implies an added effort by the authors of the article.
MATERIALS, METHODS and RESULTS:
The statistical treatment is correct.
DISCUSSION and CONCLUSION:
The conclusions are well drawn and interesting.
The discussion is correct.
Reviewer 3 Report
Comments and Suggestions for Authors
Thank you for addressing my prior concerns. I still believe that one hour is not an ideal metric, however, that was your chosen driver and as the study is written it is acceptable for publication.
